# Epidemiological Features of the Bladder Neck Rest Position and Mobility

**DOI:** 10.3390/jcm9082413

**Published:** 2020-07-28

**Authors:** Edyta Horosz, Andrzej Pomian, Aneta Zwierzchowska, Wojciech Lisik, Wojciech Majkusiak, Paweł Tomasik, Beata Rutkowska, Joana Skalska, Małgorzata Siemion, Dominika Banasiuk, Ewa Barcz

**Affiliations:** 1Department of Obstetrics and Gynecology, Multidisciplinary Hospital Warsaw-Miedzylesie, 04-749 Warsaw, Poland; edytahorosz@o2.pl (E.H.); apomian@gmail.com (A.P.); a.j.zwierzchowska@gmail.com (A.Z.); wmajkusiak@gmail.com (W.M.); p_tomasik@wp.pl (P.T.); beata_zak@interia.pl (B.R.); joanajermak@gmail.com (J.S.); siemion.malgorzata@gmail.com (M.S.); dominika.ochedalska@wp.pl (D.B.); 2Department of General and Transplantation Surgery, Medical University of Warsaw, 00-001 Warsaw, Poland; wojciech.lisik@wum.edu.pl

**Keywords:** bladder neck mobility, urethral mobility, ultrasound, female urethra

## Abstract

The data concerning epidemiological determinants of the bladder neck (BN) mobility are scarce. The aim of the study was to determine epidemiological features and identify factors influencing BN position at rest and BN mobility in patients without pelvic organ prolapse (POP). Seven hundred and ninety-six patients that attended two outpatient clinics were enrolled in the study. Position and mobility of the BN were measured with the use of pelvic floor ultrasound. Demographic and functional factors that were hypothesized to influence BN mobility were assessed. Vaginal deliveries (VDs) and age ≥65 were associated with lower BN position at rest. Higher BN mobility was observed in women with stress urinary incontinence (SUI). In obese women, higher BN position and lower BN mobility was observed compared to non-obese women, and it was correlated with longer urethras in this group of patients. VDs and their number were associated with increased BN mobility, independently of body mass index (BMI). To conclude, obesity, VDs, and age are factors associated with changes in bladder neck position at rest and its mobility. Higher BMI correlates with restricted BN mobility, and, therefore, the incidence of SUI in obese patients is probably not connected to BN hypermobility.

## 1. Introduction

The female urethra is a part of the urinary tract passing from the internal orifice at the bladder neck (BN) to external urethral orifice in the vaginal vestibule. Urethral length ranges from 19 to 51 mm, the mean length equaling 30 mm. Approximately 30% of the female population is characterized by a statistically significantly shorter or longer urethra than the average 30 mm [1].Tubular structure of the urethra is a histological continuation of the vesical trigone and is composed of two smooth muscle layers that are present throughout the upper four-fifths of the urethra: an inner thick longitudinal smooth muscle layer and an outer thin circular smooth muscle layer [2]. The external striated muscle part of the urethra is called the external urethral sphincter. It extends from the neck to the perineal membrane and accounts for 20–80% of the total urethral length, with the thickest part in the middle third of the urethra, the high-pressure zone [3]. The BN consists of the distal portion of the detrusor (including the trigone) and continues into the proximal urethra [2].

The functional anatomy of the BN and urethra has been studied extensively because of the belief that it is important for urinary continence. To date, the pathophysiology of stress urinary incontinence (SUI) has not been fully explained and the gaps in the understanding of causal mechanisms limit progress both in preventive strategies and treatment. A number of factors seem to contribute to the etiology of this condition, involving pathologies of the urethral sphincter function, bladder neck, urethra and pelvic floor muscles. However, the predominant factor influencing the occurrence of SUI remains to be elucidated [4,5]. Increased BN mobility is regarded to have an association with the occurrence of SUI [6]. It has been suggested that this results from disturbed BN and urethral support [7,8], which can be associated with pregnancy and vaginal birth [9]. During the last few decades, a variety of methods have been used to assess urethra-vesical mobility, including the Q-tip test [10] and radiographic techniques [11], as well as ultrasound techniques-introital [12], vaginal [13] or perineal [14] and magnetic resonance imaging [15]. BN descent has been postulated to be the most appropriate measure of urethral mobility [16]. It can be evaluated with the use of ultrasound examination: a non-invasive, simple and reproducible method that allows for dynamic assessment of pelvic organs. To date, several studies evaluating changes in BN mobility during pregnancy, after childbirth and its correlation to urinary incontinence have been published [9,17,18]. Still, BN hypermobility has also been reported in asymptomatic nonpregnant nulliparous women [19,20], as well as patients with pelvic organ prolapse (POP), both with and without any symptoms. Although the functional anatomy of the BN has been studied extensively, the available data concerning epidemiological determinants of urethro-vesical mobility are scarce.

The aim of the study was to determine epidemiological features and identify the independent factors influencing BN position and mobility in patients without POP.

## 2. Materials and Methods

The study was performed during the years 2017–2020. A cohort of 796 patients attending the outpatient clinic of the Department of Obstetrics and Gynecology in Multidisciplinary Hospital Warsaw-Miedzylesie and the outpatient bariatric clinic of the Department of General and Transplantation Surgery, Medical University of Warsaw were recruited to the study. The exclusion criteria were: previous prolapse and incontinence surgery, coexisting pelvic organ prolapse (POPQ > 1).

Various factors that were hypothesized to influence BN mobility were assessed: age, body mass index (BMI), height, parity, mode of delivery, birth weight of children, and age at the time of first delivery.

All women underwent pelvic floor ultrasound examination. Urethral length and BN position were measured in a standardized manner, with the patient on the gynecological chair in a semi-sitting position with the bladder filled to 200–400 mL. Three diameters of the bladder were measured in order to estimate the bladder volume before the examination. The probe (a 3.6–8.3 MHz vaginal transducer with a beam angle of 160°) was placed in the vaginal introitus at the level of the external urethral orifice. With the probe in this position, the urinary bladder, urethra, and pubic symphysis with the interpubic disc were visualized in the median sagittal plane, according to Interdisciplinary S2k Guideline: Sonography in Urogynecology [21]. The length of the hypoechogenic core of the urethra was measured from the BN to the pelvic diaphragm. This method of measurement has been proven to be highly repeatable and reproducible [22]. BN position at rest was measured as the shortest distance between the urethral-bladder junction and the horizontal line running through the lower edge of symphysis pubis and was shown in millimeters. Accordingly, BN position was measured in the maximal descent during the Valsalva maneuver (VM) (Figure 1).

BN mobility was defined as the difference between its position at rest and during the VM and was shown in millimeters.

The study was approved by the ethics committee of Medical University of Warsaw (KB/150), and all study participants gave written informed consent.

Descriptive statistical analysis expressing the quantitative and categorical variables was performed with the use of R version 4.0.0 software. Normality was tested using the Lillefors and Shapiro-Wilk W tests. We associated the degree and type of non-adherence using the T test and variance analysis (ANOVA). The Pearson correlation test was used to determine correlations between quantitative variables. Multiple regression analysis was used to present multivariate relationships and to show the influence of independent variables on a dependent variable. *p*-value < 0.05 was considered statistically significant.

## 3. Results

Seven hundred and ninety-six women were enrolled in the study. The mean age of the patients was 55.2 ± 12.8 years. The mean BMI was 29.4 ± 7.0. Among 796 patients, 352 were obese (BMI ≥ 30, as defined by the World Health Organization).

Baseline characteristics of the study group are shown in Table 1.

The mean length of the urethra in the whole study group was 30.2 ± 4.3 mm. The mean position of the BN at rest was 10 ± 7.0 mm. The mean BN position during the VM was −4.8 ± 9.3 mm, whereas the mean BN was 14.9 ± 7.1 mm.

BN position at rest was analyzed in the whole study group, as well as relative to all available demographic features listed in Table 1. It was demonstrated that BN position was statistically significantly lower in non-obese women compared to the obese. VDs and older age were associated with lower BN position in the whole study group. As obesity is often connected with pelvic floor disorders, we performed the comparative analysis of obese and non-obese women and we showed that in both groups VDs and age over 65 years were associated with lower BN position at rest (Table 2).

It can also be seen that the BN position during the VM was highly statistically significantly higher in obese women compared to women with normal body weight (*p* < 0.001). On the contrary, both VDs and age were connected with lower BN position during the VM.

BN mobility was lower in obese patients compared to non-obese women. Meanwhile, vaginal birth was associated with increased BN mobility when compared to nulliparae.

In the subgroup of obese women, VDs were associated with increased BN mobility, whereas, in the non-obese patients, after VDs, the increase of BN mobility was not statistically significant. Age did not influence BN mobility in obese patients. Meanwhile, in the subgroup of non-obese patients BN mobility was lower in older patients but the difference was not highly significant (Table 2).

Since we previously showed that urethral length is associated with BMI [1], an analysis of correlations between BMI and urethral length and BN mobility was performed. A highly statistically significant negative correlation was observed. Lower BN mobility is associated with increased BMI and urethral length (Figure 2).

Since increased BN mobility has been acknowledged as one of the risk factors of stress urinary incontinence (SUI), BN position, both at rest and during the VM, as well as BN mobility, were compared in women that required surgical treatment for SUI and women without SUI or presenting with sporadic leak, not requiring anti-incontinence procedures. A lower BN position at rest and during the VM was observed in women with SUI. BN mobility was statistically significantly higher in women with SUI compared to women that were continent (Table 3).

Taking into consideration the results listed above, correlations between BN mobility and recognized risk factors for SUI and POP were investigated.

A highly statistically significant positive correlation between the number of VDs and increased BN mobility was observed (Figure 3).

Associations between BN mobility and the number of VDs were also assessed relative to BMI. Both in the subgroup of women with BMI < 30 and in the subgroup of obese women, the number of vaginal deliveries correlated positively with BN mobility. However, in the obese group, VDs were associated with a significantly greater increase in BN mobility compared to non-obese women (BMI < 30) (Figure 4).

BN mobility correlated negatively with BMI (*p* < 0.01) (Figure 5).

The impact of BMI on BN mobility was also analyzed separately for the subgroups of non-obese and obese women. In both subgroups, BN mobility correlated negatively with BMI. (*p* < 0.01) (Figure 6).

The correlations between BMI and BN mobility were also investigated in the subgroups of women who had had VDs, with regard to the number of VDs. It was shown that the influences of BMI and VDs on BN mobility are independent. With increased BMI, BN mobility decreased, independent of the number of VDs (*p* < 0.001 in all subgroups) (Figure 7).

No correlations between BN mobility and height, birth weight of the delivered children, maximum birth weight of the delivered children, nor age at first delivery were observed, neither in the whole study group nor in the subgroups of women with normal body weight, overweight, and obese women.

## 4. Discussion

The urethra and BN are considered a synergic unit due to the fact that their anatomy and function work co-dependently. While attempts have been made to define normal BN mobility and hypermobility, the available literature does not provide data on the relationship between the BN mobility and demographic characteristics of patients. Excessive urethral mobility may be observed in women who do not suffer from any pelvic floor disorder. However, a relationship between BN hypermobility and the occurrence SUI has been reported [4,6]. The present study confirms these findings, since greater BN mobility was shown in patients requiring surgery for SUI compared to women without SUI.

Taking into consideration the literature data, as well as our own observations, we have made an attempt to evaluate demographic factors influencing the BN position and mobility, including the most important factors connected with SUI, such as VDs, age, obesity, birth weight of children, and age during first delivery. Since several studies repeatedly demonstrated an increased prevalence of pelvic floor disorder (PFD) among obese women [23], obesity was analyzed as an independent risk factor.

Compared to perineal or suprapubic approach, ultrasound assessment of the urethra by introital technique proves to be a safe, rapid and efficacious method that can be applied also to evaluate other urethral diseases, e.g., inflammatory pathologies, such as urethral diverticula or neoplasms [24,25].

In our study, the bladder volume during the assessment was 200–400 mL. As we had previously shown, the associations between the bladder filling parameters of the urethra measured in the volume range of 200–400 mL are not statistically significant [1].

Many authors tried to distinguish the limits of normality for pelvic floor ultrasound parameters of the BN mobility. Peschers et al. examined 39 women aged 18–36. BN mobility varied from 4 to 32 mm during coughing and from 2 to 31 mm during the VM [20], whereas Reed et al. demonstrated lower BN mobility, i.e., between 0 and 18.7 mm (mean 6.3 mm) in 48 asymptomatic non-pregnant nulliparous volunteers [19]. By contrast, another study of one 118 young nulligravid women showed a wide range of BN descent from 1.2 to 40.2 mm (mean 17.4 mm) [26]. The same group of authors described 178 nulliparous Caucasian female twins and their sisters, suggesting significant congenital contribution to the phenotype of BN mobility, with approximately 50% of variability due to genetic factors in nulligravid women. [27]. Naranjo-Ortiz et al. analyzed functional BN anatomy in a retrospective study in which 429 women that underwent urodynamic testing were enrolled. Considering its association with SUI, the authors suggested a cutoff of 25 mm for BN descent to define as abnormal [16]. In the current study, we analyzed BN position and mobility in a group of 796 patients. To our knowledge, it is the largest analysis published to date. BN position at rest was assessed in all women. Demographic factors potentially associated with this parameter were also investigated. Both age and VDs were associated with lower BN position at rest. The mean BN position in the whole group was 10.0 ± 7.0 mm. In an analysis performed by Jundt et al., no significant differences between the BN position before and after delivery were observed in one hundred twelve primiparous women [6]. In our study, the position of BN was significantly lower in the group of women who delivered vaginally than in nulliparous patients. The discrepancy may result from a larger sample size in our study and the fact that in the above cited study sonographic examination was performed a few months after delivery, whereas in the current analysis patients with a history of VD were included regardless to the time that passed after their deliveries.

In the current analysis, we showed that BN position at rest decreases with age and is lower in women aged 65 or over compared to younger patients. This is in agreement with other authors, who indicated age as a risk factor for both POP and SUI [28].

We also demonstrated that in obese patients the position of the BN at rest is higher than in non-obese women. We had previously reported that this phenomenon is connected with significantly longer urethras in obese patients [1].

In order to assess BN mobility in the analyzed population, we took into consideration demographic factors that may affect this parameter, i.e., number of deliveries, age, BMI, height, and medium and maximum birth weight of children, as well as age during the first delivery. Only the first three factors were proven to influence BN mobility.

Pregnancy and childbirth are risk factors for SUI. The occurrence of SUI is higher after VD than after cesarean section [29]. This may be the result of congenital factors, hormonal changes, or neuromuscular damage to the urethral support structures. Most studies investigated changes of the position and mobility of the urethro-vesical junction during the third trimester of pregnancy and several weeks to six months postpartum in relation to incontinence. Toozs-Hobson et al. studied 110 primigravid women recruited between 32 weeks and term and who completed the 6-months’ follow-up. Compared to antenatal measurements, VD was associated with increased BN mobility and lower BN position [30]. This is with agreement with a prospective observational study performed by Dietz and Bennett, who examined 169 nulliparous women at 6–18 weeks, 32–37 weeks, and 2–5 months after childbirth. VD was associated with significantly increased mobility of the proximal urethra and BN, as well as decreased BN position. Forceps delivery was connected with the most pronounced effect [31]. One of the longest observations assessing BN and urethral mobility from pregnancy to 4 years postpartum included 180 women and confirmed the abovementioned associations between vaginal delivery and BN descent and mobility. Again, operative vaginal deliveries proved to have the strongest correlation with hypermobility of these structures. The authors reported increased values for all ultrasound variables over time, especially during the interval between 1 and 4 years after delivery [9].

In the present study, we confirmed that vaginal childbirths increase BN mobility and that BN mobility was positively correlated with the number of deliveries. This outcome was independent of BMI. Our results are consistent with those obtained by authors cited above. Large sample size and the fact that obesity was analyzed as an independent risk factor may be considered the strengths of the current study.

Despite the fact that women aged 65 or over had a lower BN position, BN mobility did not increase with age. According to the literature, the risk of SUI is positively correlated with age [28,32]. Our results suggest that this phenomenon may not be a result of increased BN mobility in older women.

In the current study, urethral mobility decreased with increasing BMI, and this effect was observed in the whole study group, both women with normal weight and obese patients. This is certainly related to significantly longer urethras [1] and higher BN position at rest in obese patients shown in the current study. It is widely agreed that obesity influences various kinds of lower urinary tract symptoms, including SUI. Associations between BMI and urinary incontinence were evaluated in several studies [33,34]. The results of the present study implicate that the higher occurrence of SUI in obese patients is not related to BN mobility. Presumably, urethral sphincter deficiency and the higher intra-abdominal pressure are the primary factors responsible for stress incontinence in those women.

## 5. Conclusions

VDs and age over 65 were identified as demographic factors associated with lower BN position at rest. On the contrary, obesity is associated with higher BN position, and it is dependent on the longer urethra in these women as compared to non-obese women. 

Higher mobility of the BN was observed in women requiring medical intervention because of SUI as compared to continent patients, confirming importance of this factor in the pathophysiology of SUI. We also showed that VDs and their number positively correlated with BN mobility and that the phenomenon was independent of BMI, which confirms delivery-dependent urethral hypermobility. We demonstrated that obesity is correlated with restriction of the BN mobility and strongly connected to urethral length. These findings may suggest that higher incidence of SUI in obese patients is not associated with hypermobility of BN.

## Figures and Tables

**Figure 1 jcm-09-02413-f001:**
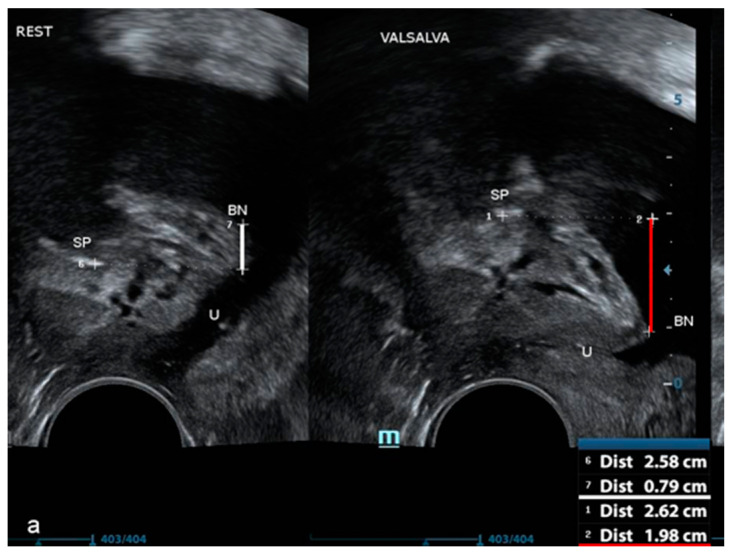
The technique of bladder neck (BN) position measurement.

**Figure 2 jcm-09-02413-f002:**
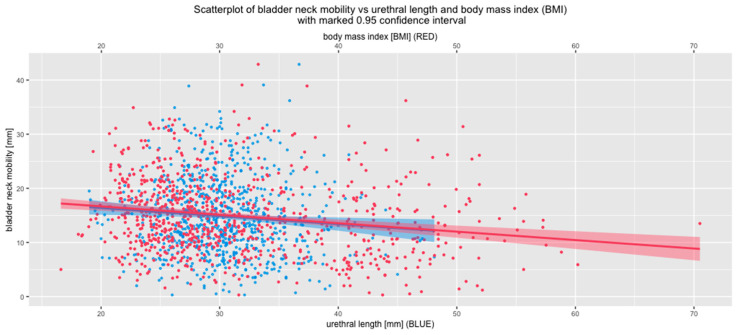
Associations between bladder neck mobility, urethral length, and body mass index (BMI).

**Figure 3 jcm-09-02413-f003:**
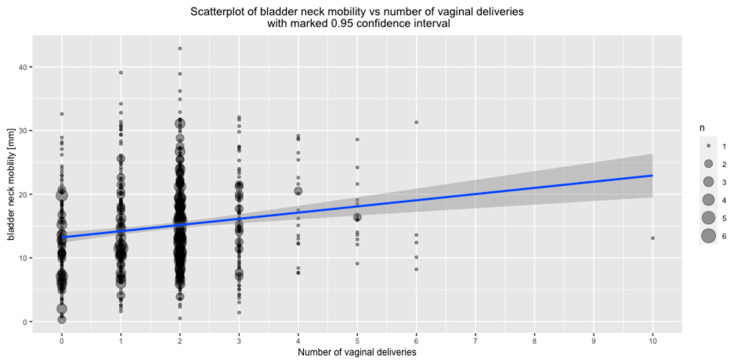
Correlation between bladder neck mobility and number of vaginal deliveries.

**Figure 4 jcm-09-02413-f004:**
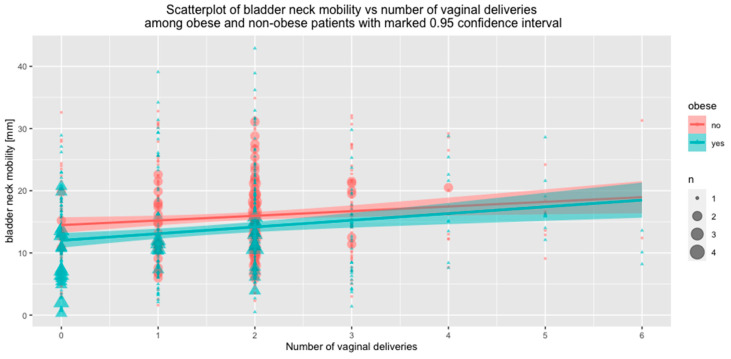
Correlation between the number of vaginal deliveries and bladder neck mobility in non-obese and obese women (*p* < 0.001).

**Figure 5 jcm-09-02413-f005:**
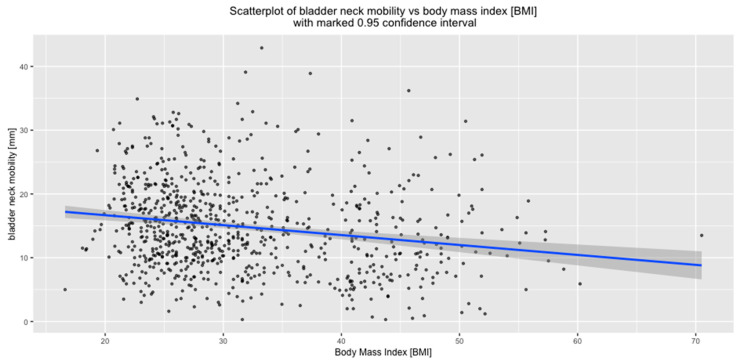
Correlation between BN mobility and BMI.

**Figure 6 jcm-09-02413-f006:**
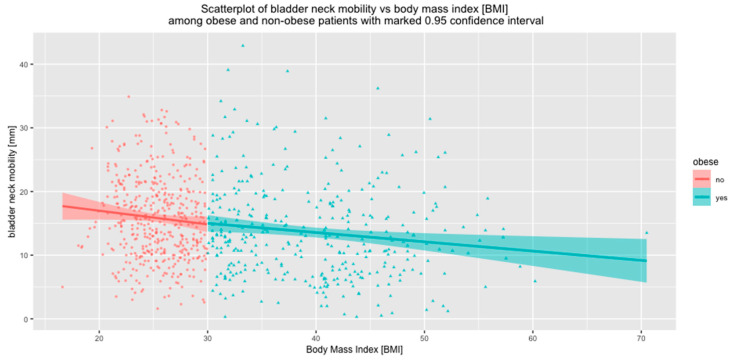
Correlation between BN mobility and BMI in subgroups of non-obese and obese women.

**Figure 7 jcm-09-02413-f007:**
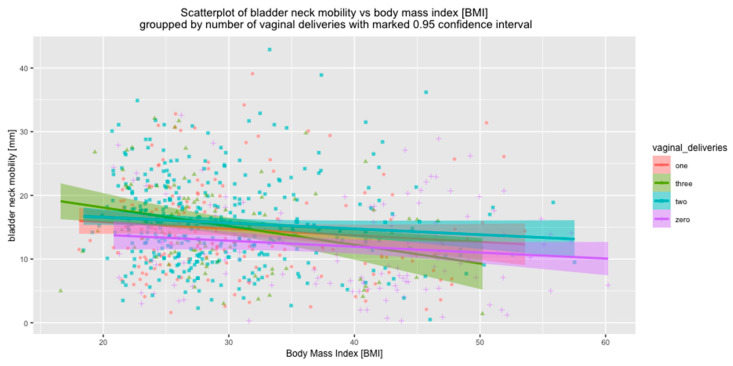
Correlation between bladder neck mobility and BMI, relative to the number of vaginal deliveries.

**Table 1 jcm-09-02413-t001:** Cohort description and analyzed demographic features (*n* = 796).

Variable	Mean	Minimum	Maximum	Standard Deviation
Age (years)	53.4	18.0	85.0	13.3
Height (cm)	164.2	145.0	190.0	6.1
Weight (kg)	85.6	44.0	176.0	25.3
BMI	31.7	16.6	70.5	8.8
No of deliveries	1.9	0.0	10.0	1.1
No of VDs	1.7	0.0	10.0	1.2
No of cesarean sections	0.2	0.0	3.0	0.5
No of vacuum/forceps deliveries	0.0	0.0	2.0	0.1
Average birth weight (VDs only) (g)	3432	650	5050	500
Maximum birth weight (VDs only) (g)	3602	650	5850	555
Cumulative birth weight (VDs only) (g)	7045	650	30,800	3460
Age at 1st VD (years)	23.9	14.0	40.0	3.9
Age at 2nd VD (years)	27.5	15.0	43.0	4.6
Age at last VD (years)	28.6	17.0	46.0	5.3
Urethral length (mm)	30.2	19.0	48.1	4.3
BN position at rest (mm)	10.0	−31.8	29.9	7.0
BN position during the VM	−4.8	−37.1	24.8	9.3
BN mobility (rest–VM) (mm)	14.9	0.3	42.9	7.1

BMI–body mass index; BN—bladder neck; VD—vaginal delivery; VM—Valsalva maneuver.

**Table 2 jcm-09-02413-t002:** Bladder neck position at rest, during the Valsalva maneuver, and bladder neck mobility in examined groups of patients.

	BN	At Rest [mm]	During the VM [mm]	BN Mobility (rest-VM) [mm]
Position Patients				
Whole Group *n* = 796		10.0 ± 7.0	−4.8 ± 9.3	14.9 ± 7.1
Obese *n* = 352		11.7 ± 6.8	−2.0 ± 9.6	13.6 ± 7.4
Non-Obese *n* = 444		8.7 ± 6.9	−7.1 ± 8.5	15.8 ± 6.6
*p*		<0.001	<0.001	<0.001
No of VDs = 0 *n* = 146		13.4 ± 6.0	1.3 ± 8.5	12.1 ± 6.7
No of VDs > 0 *n* = 650		9.2 ± 7.0	−6.2 ± 8.9	15.5 ± 7.0
*p*		<0.001	<0.001	<0.001
<65 years *n* = 613		11.1 ± 6.4	−4.1 ± 9.2	15.1 ± 7.2
≥65 years *n* = 182		6.4 ± 7.7	−7.4 ± 9.1	13.8 ± 6.3
*p*		<0.001	<0.001	<0.05
Obese with VD = 0 *n* = 102		14.5 ± 5.9	3.3 ± 7.8	11.2 ± 6.3
Obese with VD > 0 *n* = 250		10.5 ± 6.8	−4.1 ± 9.3	14.6 ± 7.6
*p*		<0.001	<0.001	<0.001
Non-obese with VD = 0 *n* = 44		10.8 ± 5.4	−3.4 ± 8.0	14.2 ± 7.2
Non-obese with VD > 0 *n* = 400		8.5 ± 7.0	−7.6 ± 8.4	16 ± 6.5
*p*		<0.05	<0.01	Ns
Obese < 65 years *n* = 281		12.8 ± 6.3	−1.1 ± 9.6	13.9 ± 7.8
Obese ≥ 65 years *n* = 71		7.0 ± 6.8	−5.5 ± 8.1	12.5 ± 5.5
*p*		<0.001	<0.001	Ns
Non-obese < 65 years *n* = 332		9.6 ± 6.2	−6.6 ± 8.1	16.2 ± 6.5
Non-obese ≥ 65 years *n* = 111		6.1 ± 8.2	−8.6 ± 9.4	14.7 ± 6.6
*p*		<0.001	<0.05	<0.05

Data are given as mean, ± SD.; BN—bladder neck; VD—vaginal delivery; VM—Valsalva maneuver; ns—not significant.

**Table 3 jcm-09-02413-t003:** Bladder neck position at rest, during the Valsalva maneuver, and bladder neck mobility in patient with stress urinary incontinence (SUI) that required anti-incontinence procedures compared to women without SUI or with minor incontinence not requiring surgical intervention.

	BN Position	At Rest [mm]	During the VM [mm]	BN Mobility (Rest-VM) [mm]
Patients				
Not scheduled for anti-incontinence surgery *n* = 426		11.1 ± 7.7	−2.7 ± 10.3	13.8 ± 7.2
Scheduled for anti-incontinence surgery *n* = 370		8.7 ± 5.8	−7.4 ± 7.2	16.1 ± 6.8
*p*		*p* < 0.001	*p* < 0.001	*p* < 0.001

Data are given as mean, ± SD; BN—bladder neck; VM—Valsalva maneuver.

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
