# Peer review of "Epidemiological Features of the Bladder Neck Rest Position and Mobility"

_jcm, 2020, doi:10.3390/jcm9082413_

Round 1
Reviewer 1 Report
The paper is very good and interesting, well written and clear. However I suggest only few comments to improve the paper:
1) Measurement of BN descent by US in the method section
You should insert at least one Reference (or this is a new method ?). In particular you should compare this method with those used by authors cited in the discussion.
Which US probe did you use ? I guess an end-fire o endovaginal convex probe.
2) The mean length of the urethra in the whole study group was 30.2±4.3 mm. The mean position of the BN at rest was 10± 7.0 mm. The mean BN position during the VM was -4.8± 9.3 mm, whereas the mean urethral mobility was 14.9±7.1 mm. I suggest to modify the last sentence: …, whereas the mean BN mobility was 14.9±7.1 mm.
3) add in the discussion section: Ultrasound assessment of the urethra by introital technique is safe, rapid and efficacy in order to evaluate other urethral diseases if compared to perineal or sovrapubic approach. Introital US may shows easily associated disease such as inflammatory disease such as urethral diverticula or neoplasms [ Dell'Atti L, Galosi AB. Female Urethra Adenocarcinoma. Clin Genitourin Cancer. 2018;16(2):e263-267. Galosi AB, Dell’Atti L. Ultrasound Study of the Urethra. Chapter 18 Pag 211-226. In Book: Atlas of Ultrasonography in Urology, Andrology, and Nephrology. P. Martino, A.B. Galosi (eds.) Springer International Publishing Switzerland 2017.]
Author Response
Thank you very much for taking the time to review our article and suggesting valuable improvements to the manuscript.
All changes made to the manuscript are in red.
Ad. 1. The paragraph in the Materials and Methods section describing pelvic floor ultrasound was extended in order to give detailed information on the method. Two references were also added.
Ad. 2. The suggested modification has been made.
Ad. 3. The suggested paragraph was added to the text in the Discussion section.
Reviewer 2 Report
This manuscript shows the bladder neck rest position and mobility using transperineal US. This study is interesting, and the inclusion number of this study is enough (796 patients). However, there are some issues to be addressed.
Bladder neck’s position and mobility can be assessed by transperineal US with a high degree of reliability. However, limitation is related to the transperineal US technique, the author should be discussed this point in discussion section.
How about the bladder volume? Full and empty bladder might be influencing the results of this study. The author describes this point in the section of methods and discussion section.
Author Response
Thank you very much for taking the time to review our article and suggesting valuable improvements to the manuscript.
All changes made to the manuscript are in red.
Ad. 1. Both transperineal and introitus US techniques have been demonstrated to be adequate and reproducible methods to assess pelvic floor structures. Pelvic floor ultrasound (PF US) technique combines the advantages of both these approaches [Tunn R, Albrich S, Beilecke K, et al. Interdisciplinary S2k guideline: sonography in urogynecology: short Version—AWMF registry number: 015/055. Geburtshilfe Frauenheilkd. 2014;74:1093–1098]. This method of measurement has been proven to be highly repeatable and reproducible. [Wlazlak E, Kociszewski J, Suzin J, Dresler M, Surkont G. Urethral length measurement in women during sonographic urethrocystography—an analysis of repeatability and reproducibility. J Ultrason. 2016;16:25–31] (this information has been added to the Materials and Methods section). However, the aim of the current study was to observe the changes in BN position depending on various factors, not to assess the applied technique itself.
Ad.2. The bladder filling was 200-400 ml. Adequate information was added to the Materials and Methods section and the Discussion section.